# Spatial Configurations of 3D Extracellular Matrix Collagen Density and Anisotropy Simultaneously Guide Angiogenesis

**Steven A. LaBelle**[1,2], **A. Marsh Poulson IV**[2], **Steve A. Maas**[1,2], **Adam Rauff**[1,2], **Gerard A. Ateshian**[3], **Jeffrey A. Weiss**[1,2]*

**1** Department of Biomedical Engineering, University of Utah, Salt Lake City, Utah, United States of America, **2** Scientific Computing and Imaging Institute, University of Utah, Salt Lake City, Utah, United States of America, **3** Department of Mechanical Engineering, Columbia University, New York, New York, United States of America

* jeff.weiss@utah.edu

**Data Availability Statement:** Data and code are available at github.com/febiosoftware/AngioFE. The current source code is presented in the head directory. Source code, binary files, and input files

## Abstract

Extracellular matrix (ECM) collagen density and fibril anisotropy are thought to affect the development of new vasculatures during pathologic and homeostatic angiogenesis. Computational simulation is emerging as a tool to investigate the role of matrix structural configurations on cell guidance. However, prior computational models have only considered the orientation of collagen as a model input. Recent experimental evidence indicates that cell guidance is simultaneously influenced by the direction and intensity of alignment (i.e., degree of anisotropy) as well as the local collagen density. The objective of this study was to explore the role of ECM collagen anisotropy and density during sprouting angiogenesis through simulation in the AngioFE and FEBio modeling frameworks. AngioFE is a plugin for FEBio (Finite Elements for Biomechanics) that simulates cell-matrix interactions during sprouting angiogenesis. We extended AngioFE to represent ECM collagen as deformable 3D ellipsoidal fibril distributions (EFDs). The rate and direction of microvessel growth were modified to depend simultaneously on the ECM collagen anisotropy (orientation and degree of anisotropy) and density. The sensitivity of growing neovessels to these stimuli was adjusted so that AngioFE could reproduce the growth and guidance observed in experiments where microvessels were cultured in collagen gels of varying anisotropy and density. We then compared outcomes from simulations using EFDs to simulations that used AngioFE's prior vector field representation of collagen anisotropy. We found that EFD simulations were more accurate than vector field simulations in predicting experimentally observed microvessel guidance. Predictive simulations demonstrated the ability of anisotropy gradients to recruit microvessels across short and long distances relevant to wound healing. Further, simulations predicted that collagen alignment could enable microvessels to overcome dense tissue interfaces such as tumor-associated collagen structures (TACS) found in desmoplasia and tumor-stroma interfaces. This approach can be generalized to other mechanobiological relationships during cell guidance phenomena in computational settings.

used for the simulations and analyses in this study can be found in the documentation directory of the GitHub repository.

**Funding:** Financial support from the National Institutes of Health (www.nih.gov) is gratefully acknowledged: NIH R01GM083925 (JAW, GAA), NIH 5R01HL131856 (JAW), NIH U24EB029007 (JAW), NIH R01AR069297 (JW). The funders had no role in study design, data collection and analysis, decision to publish, or preparation of the manuscript.

**Competing interests:** The authors have no competing interests to declare.

## Author summary

Matrix collagen fibril anisotropy and density are gaining recognition for their mechanoregulatory roles in cellular growth and guidance. For instance, we recently demonstrated that new vessel growth increases with the degree of matrix collagen alignment in a density-dependent manner during *in vitro* angiogenesis. The spatial configuration of collagen fibril alignment and density between adjacent tissues is thought to affect development of new vasculatures during pathologic and homeostatic angiogenesis. Computational simulation is emerging as a tool to evaluate how the integrated effects of different matrix structural cues are involved in guidance and deflection of microvessels between tissues. Thus, the objective of this study was to incorporate our prior experimental finding that angiogenic neovessels are simultaneously sensitive to ECM collagen density and fibril anisotropy (orientation and degree of anisotropy). We found that fibril anisotropy emboldens neovessels to migrate long distances and persist through dense tissue interfaces. These findings have implications for angiogenesis during wound healing and pathologies such as cancer tumorigenesis.

## Introduction

Structural properties of the extracellular matrix (ECM)–namely collagen fibril anisotropy and collagen density–regulate cellular proliferation and directional guidance (taxis) [1–3]. During neovascularization, for instance, both homeostatic (wound healing, implant inosculation) and pathologic (tumorigenesis, chronic inflammation, arthritis) angiogenesis are associated with inhomogeneous matrix structures that can encourage or inhibit vascular growth [4–7]. At the cellular level, tissue structure directly influences cellular proliferation and cytokine signaling kinetics [1, 8, 9]. Experimental methods to delineate the role of individual ECM structural features on cellular proliferation and directional guidance remain impractical or impossible. Thus, computational simulation is emerging as a tool to investigate the influence of continuum-based measures of ECM collagen microstructure on cell guidance.

The mass density of collagen fibrils and their anisotropy in the ECM play unique roles in modulating growth rate during angiogenesis and guiding growing neovessels [2, 8]. Collagen density regulates ECM stiffness and thus contributes to durotaxis–the guidance of cells by rigidity gradients [10]. Fibril anisotropy can be described by the fractional density of collagen fibrils oriented along all spatial directions emanating from a material point. This anisotropy contributes to durotaxis via guidance of cells along variously-tensed fibrils, as well as contact guidance and haptotaxis–the guidance of cells by matrix binding site gradients (e.g., integrins) [11, 12]. In a recent study, we demonstrated that both microvessel rate of extension and guidance (reorientation and taxis) increased with matrix anisotropy during *in vitro* microvascular angiogenesis [1]. Interestingly, collagen density attenuated the effects of anisotropy so that stronger alignment was required for significant proangiogenic effects to occur in dense scaffolds. The specific configuration of collagen density gradients and fibril alignment are thought to contribute to the development of new vasculatures during pathologic and homeostatic angiogenesis [13, 14]. However, it remains difficult (or impossible) to fabricate scaffolds that adequately and exhaustively represent complex collagen matrix architectures such as granulation tissue or along tissue-tissue interfaces. *In vivo* studies present additional challenges due to the number of confounding immunoregulatory, patient-specific, and etiology-specific factors [15–17]. Thus, simulation can be used to study the isolated effects and coordination of

multiple stimuli in biomedical problems. Further, simulation can identify specific hypotheses and narrow down experimental parameter spaces. Toward this end, we present a computational simulation framework that predicts the effects of different configurations of collagen density and fibril anisotropy during angiogenesis.

Several approaches have been employed to simulate fibril guidance during angiogenesis and other cell migration phenomena. Vector fields have long been used to represent spatial variations in collagen alignment at the sub-millimeter scale [18–22]. Here, the predominant orientation of the collagen matrix at each spatial discretization is represented by a unit vector. In a similar approach, discrete fibril networks have been superimposed on the simulation domain so that cells encounter multiple fibrils of differing orientation [5]. In both cases, cells have been modeled as discrete agents that migrate along the direction of the locally interpolated vector or embedded fibrils. However, cells are substantially larger than fibrils, and they interact with many fibrils of varying orientation. Additionally, these simple approaches are insufficient to determine the degree of anisotropy.

Recently, orientation distribution functions (ODFs) have become a popular alternative for describing fibril anisotropy since they summarize the orientations of countless fibrils at a point in a continuum [11, 23–27]. More generally, ODFs characterize diverse types of oriented data including fibril alignment, trajectories of migrating cells, and entire microvascular networks [7, 8, 13, 20]. The values of the ODF correspond to the relative probability of observations (fibrils, cell trajectories, vessels) oriented along each direction. Mathematically, the ODF can be defined as $\Omega : \mathbb{S}^{n-1} \to \mathbb{R}^1_+$, where $\mathbb{S}^{n-1}$ is the space of all unit vector orientations $\boldsymbol{u} \in \mathbb{R}^n$ originating from the ($n$-1)-sphere, and $\mathbb{R}^1_+$ is the space of all real, positive scalars [28]. The ODF is often formulated as a probability density function with the constraint that integration over the ODF sums to 1:

$$\int_{\mathbb{S}^{n-1}} \Omega(\boldsymbol{u}) d\mathbb{S}^{n-1} = 1 \qquad [1]$$

For most practical cases, integration takes place along the circumference of a circle (2D distributed data, $n = 2$) or over the surface a sphere (3D distributed data, $n = 3$). ODFs are commonly parameterized as ellipsoidal fibril distributions (EFDs) or periodic von-Mises distributions based on the shape of their boundary [11, 27]. Matrix-valued tensors (structure tensors) can be formulated that encode the shape, orientation, and anisotropy of the parameterized ODF.

Unlike vector field models, ODF-based models typically model cell populations as continuous concentrations whose migration (modeled as anisotropic diffusion) is influenced by the ODF shape and orientation. Such models, however, have generally neglected the degree of anisotropy as a model input, despite evidence that anisotropy significantly affects neovessel extension rate, proliferation, guidance, and persistence [1, 8, 29]. Furthermore, the assumption that cell populations move as a continuum is ill-suited for phenomena characterized by stochastic behaviors of individual migrating cells (e.g., neovessel recruitment by solid tumors, circulating tumor cell extrusion/extravasation) [30, 31]. This is particularly relevant for simulations of microvascular angiogenesis, which model the formation and extension of discrete endothelial cell networks that interact with each other and the ECM. In the case of discrete cell modeling, vector fields have been preferred over ODFs because vectors can easily be interpolated and they identify a single direction for each cell agent to grow. In contrast, ODFs require advanced calculations to interpolate, and the determination of which direction each cell agent migrates is non-intuitive.

The objectives of this study were 1) to develop an enhanced simulation framework to predict microvascular growth and guidance based on 3D EFD representations of matrix collagen

anisotropy and density, and 2) to apply the framework to predict experimental observations of microvascular growth in the contexts of wound healing and tumorigenesis. To do this, we extended our AngioFE simulation framework to integrate prior experimental findings that the degree of matrix anisotropy and density simultaneously affect microvascular growth [1]. First, we simulated anisotropic fibril guidance during sprouting angiogenesis where matrix anisotropy was described using a field of either vectors or EFDs. We parameterized collagen fibril ODFs as EFDs due to their simplicity, as well as their agreement with experimental ODFs extracted from images of collagen hydrogels. Simulation results were compared to experimental measures of microvascular growth and guidance in anisotropic matrices of differing densities. Relationships between the local matrix structure (collagen density, anisotropy) and the growth behavior (rate, orientation) were updated in AngioFE to reflect experimental findings. Additionally, we evaluated the accuracy of pseudo-deformed structure tensors using techniques from differential geometry. Second, we investigated the role of physiologically relevant configurations of anisotropy and density such as anisotropy gradients in soft tissues during wound healing as well as structural alterations of the stroma during cancer tumorigenesis. The presented 3D simulation framework is the first to include matrix density and the degree of anisotropy simultaneously as model inputs. Our results suggest that spatially patterned structural cues can determine the success or failure of neovascularization in physiological and pathological contexts.

## Results

### AngioFE model formulation

To address current shortcomings in modeling anisotropic guidance of angiogenesis, we extended our AngioFE simulation framework that models sprouting angiogenesis from parent microvessel fragments within the FEBio finite element software suite (Finite Elements for Biomechanics, FEBioStudio, www.febio.org) [21, 32]. Briefly, microvessel networks were explicitly represented as connected line segments whose tips grew through and deformed a finite element mesh representing the surrounding tissue. Previously, AngioFE modeled fibril guidance via vector fields that described the average collagen orientation at any position in the simulation. However, this approach failed to account for the local degree of ECM anisotropy and thus tended to over-constrain vessel reorientation. Thus, we modified AngioFE to represent fibril guidance via EFD fields. Notably, each EFD encoded the degree of anisotropy. This section briefly summarizes the formulation of FEBio + AngioFE as shown in the graphical summary (Fig 1). The full algorithmic details are described in the Methods and Supporting Methods A-M in S1 Text. All code and data are available on the AngioFE GitHub repository at github.com/febiosoftware/AngioFE.

**Model inputs.** FEBioStudio, the graphical user interface for FEBio, was used to generate the geometry and finite element discretizations, assign the constitutive model and material coefficients, and prescribe boundary conditions. AngioFE parameters were provided to dictate the distribution of parent microvessel fragments and to define the rules that controlled the growth rate, growth direction, and tip cell contraction.

**Model initialization.** We initialized our models to mimic the starting density of parent microvessels in collagen hydrogel microvessel culture experiments [1]. Spatial maps of model physical parameters (e.g., EFD and density maps) were then initialized.

**Vessel orientation.** The new direction each vessel tip grew, $\psi_{new}$, was assumed to emerge from competition between persistence (growth along the vessel's current orientation, $\psi$) and growth along the direction of contact guidance ($\theta$) [2, 4]. To determine $\theta$, we first interpolated the local ODF from the integration points of the finite element containing the tip [33]. Next,

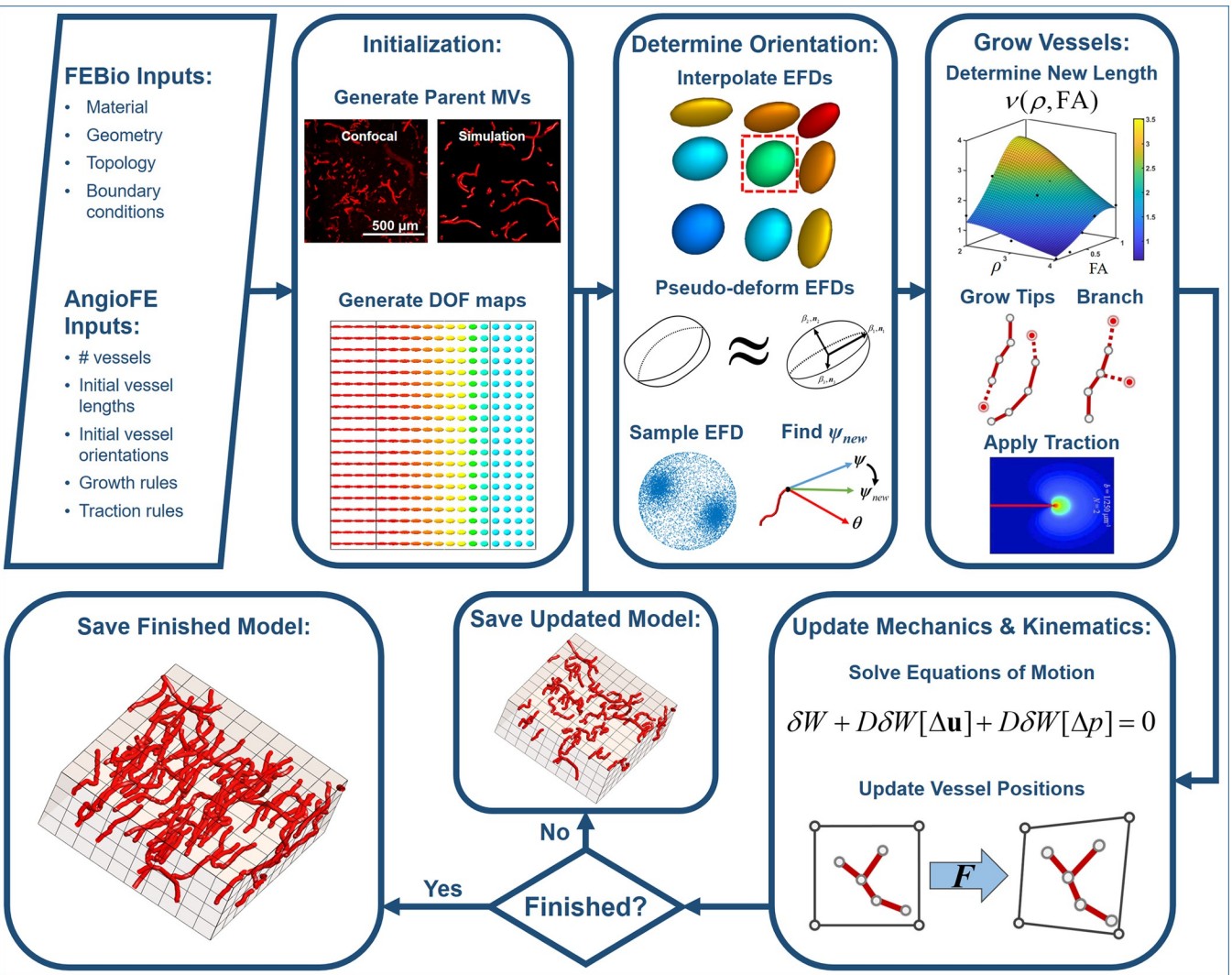

**Fig 1. FEBio + AngioFE Graphical Summary.** Presented here is a graphical summary of the simulation framework. The physical parameters of the model were generated and prescribed in FEBio/FEBioStudio. Other user-defined parameters associated with vessel growth and traction behaviors were included to configure AngioFE. **Model initialization:** Parent microvessel fragments were superimposed on the finite element mesh. Degrees of freedom (DOF) such as collagen EFDs and density were mapped to the finite element mesh. **Vessel orientation:** The direction each vessel grew, $\psi_{new}$, was determined by 1) interpolating EFDs from the finite element integration points to the vessel tip, 2) pseudo-deforming the local EFD, 3) sampling the EFD for a single contact guidance direction $\theta$, and 4) determining the balance between persistence along the previous vessel orientation $\psi$ and contact guidance by $\theta$. **Vessel extension:** The function $v(\rho,\mathrm{FA})$ scaled the vessel extension rate. This function decreased inversely with collagen density and increased directly with collagen anisotropy. After new vessels grew, branches were added along the existing vessels. Finally, cellular tractions were applied at the tips of the new vessels. **Mechanics & kinematics:** Cell tractions were sent to FEBio, which then solved the equations of motion. Vessel positions were then updated by AngioFE. **Model I/O:** The updated vessel positions and finite element degrees of freedom were saved after each time step.

we determined the pseudo-deformed EFD configuration. Pseudo-deformation is a process that identifies an EFD that closely approximates the true ODF, which is important since deformed EFDs may morph into ODFs that are not ellipsoidal [11]. To mimic filopodial probing of the ECM prior to cell elongation, we introduced a method to sample orientation vectors $\boldsymbol{\theta}$ from EFDs via Monte-Carlo methods. The direction that each vessel grew, $\psi_{new}$, was determined by partially rotating $\psi$ towards $\theta$ about their shared orthogonal axis (See Methods: "FEBio + AngioFE computational model of microvascular growth", Supporting Methods A in S1 Text and S1 Fig).

**Vessel extension rate and tractions.** The length each vessel grew during a simulation time step was determined from the vessel extension rate. The total vessel length was assumed to increase sigmoidally with respect to time based on time-series morphometric measurements of angiogenic microvessels in culture [20]. The time-dependent growth was scaled by $v(\rho,\text{FA})$, a Lorentzian function which decreased with matrix density and increased with anisotropy. New vessel tips were added at the position determined by $\psi_{new}$ and the vessel extension rate. Afterwards, branches were stochastically introduced as new vessel tips emerging from older vessel segments (Supporting Methods J in S1 Text). Finally, traction fields were imposed around new vessel tips, which allowed them to deform the surrounding ECM [19, 34].

**Updating mechanics and kinematics.** FEBio solved the equations of motion in response to cell tractions, geometric boundary constraints, viscoelasticity, and interstitial fluid motion. The deformed matrix was returned to AngioFE, which updated vessel positions via kinematics.

**Model I/O.** The solution was saved after each time step. Output data included vessel positions, vessel lengths, branch counts, finite element nodal displacements, and finite element integration point values for vessel tractions, EFDs, and matrix density. Model results were visualized in FEBioStudio and data were analyzed and post-processed in MATLAB (Math-Works, Natick, MA).

## EFD fields enhanced predictions of cell guidance and migration

To highlight the crucial role that the degree of anisotropy plays during cell guidance, we simulated microvessel growth using either vector fields or EFD fields. Simulations were generated to correspond to experiments from our recent *in vitro* study on the roles of matrix density and anisotropy during angiogenesis [1]. Microvessel culture was performed or simulated in low, medium, or high degrees of collagen alignment as well as low or high collagen density (Figs 2, 3A, and S3A). ODFs of microvascular orientation were extracted either from confocal images or from model outputs. Extracted ODFs were then projected onto the XY (Figs 3B and S3B) and XZ planes. We primarily evaluated the XY plane projections since there was minimal growth in the Z direction (as previously observed [3, 35]). There was good agreement between the experimental (confocal) microvascular ODFs and those generated by EFD simulations. ODFs associated with vector field simulations were less polarized and under-predicted microvascular reorientation for the cases of medium and high anisotropy. Further, vector field approaches under-predicted microvascular alignment in sensitivity studies even when growth was simulated to occur solely along the local collagen orientation (i.e., when the internal mechanism representing persistence was disabled; Supporting Methods A-C in S1 Text and S1 and S2 Figs). Similar trends were observed regardless of matrix density (S2 and S3 Figs). Both approaches were insensitive to finite element mesh refinement (Supporting Methods D in S1 Text and S4 Fig). Furthermore, both approaches were insensitive to experimentally established levels of vessel traction (Supporting Methods M in S1 Text and S15 and S16 Figs) [3, 19, 34]. However, we found that stronger tractions (10X experimental observations) initiated a positive feedback loop, resulting in heightened polarization of emerging vascular networks along the horizontal orientation. This feedback loop was more pronounced in vector field simulations compared to EFD simulations.

## Pseudo-deformed EFDs closely approximated 3D fiber distribution deformation

The accuracy of pseudo-deformed EFDs has not previously been calculated, despite growing adaptation of the method [11, 25, 27]. Thus, we validated and verified pseudo-deformation 1)

**Fig 2. Generation of collagen gels with varying anisotropy.** Collagen gels were aligned to low, moderate, and high anisotropy and imaged via second harmonic generation (SHG) in a prior study [1]. Image data was used to extract ODFs and fit EFDs for each level of alignment.

qualitatively by assessing the pseudo-deformation of local EFDs during tissue-level strains and 2) quantitatively by using differential geometry measures to compare ODFs after non-affine deformation ($\Omega$) to ODFs after pseudo-deformation ($\Omega^p$). Notably, pseudo-deformation assumes there is no interaction between fibrils (i.e., fibrils deform due to tissue-level strains and cell tractions, but not due to bonds between fibrils).

For qualitative analysis, the uniaxial tension test of a collagen gel was simulated to 100% elongation using FEBio (Fig 4A). Visual observation demonstrated that pseudo-deformed EFDs experienced rotation and stretch in regions characterized by symmetric or non-affine local deformations that occur at the gel edges, center, and through the thickness. We quantitatively verified our approach by calculating the difference in generalized fractional anisotropy (GFA, no units, range [0, 1]) between deformed and pseudo-deformed distributions after undergoing tension, compression, simple shear, or pure shear (Supporting Methods E in S1 Text). Differences in GFA were less than $1 \times 10^{-3}$ for all cases (Figs 4B and 4C, and S5–S8). Similarly, we measured the Fisher-Rao distance between the deformed and pseudo-deformed ODFs. The Fisher-Rao distance is a measure of dissimilarity between distributions, where the distance between identical distributions is 0° and the maximum possible distance between distributions is 90°. In our numerical simulations, the distance for all cases was below 6°, a threshold that we previously identified was explainable by random sampling [28]. Pseudo-deformed EFDs were close to the true ODFs for up to 50% applied tension/compression and 45% shear in numerical experiments regardless of the initial matrix anisotropy. These results demonstrate that pseudo-deformed EFDs ($\Omega^p$) adequately approximate non-affine deformations $\Omega$ at physiologically relevant levels of strain.

### Anisotropy gradients recruit angiogenic neovessels

Recent experimental efforts have established that endothelial cell migration and guidance are affected by the degree of tissue anisotropy. This led us to the question: "how might spatial variations in anisotropy (i.e., anisotropy gradients) affect neovessel guidance and vascular recruitment?" Thus, we generalized physiologically relevant anisotropy gradients in AngioFE. A rectilinear simulation domain was created and classified into three regions: the proximal (left), middle, and distal (right) regions (Fig 5). Parent microvessels were generated in the proximal

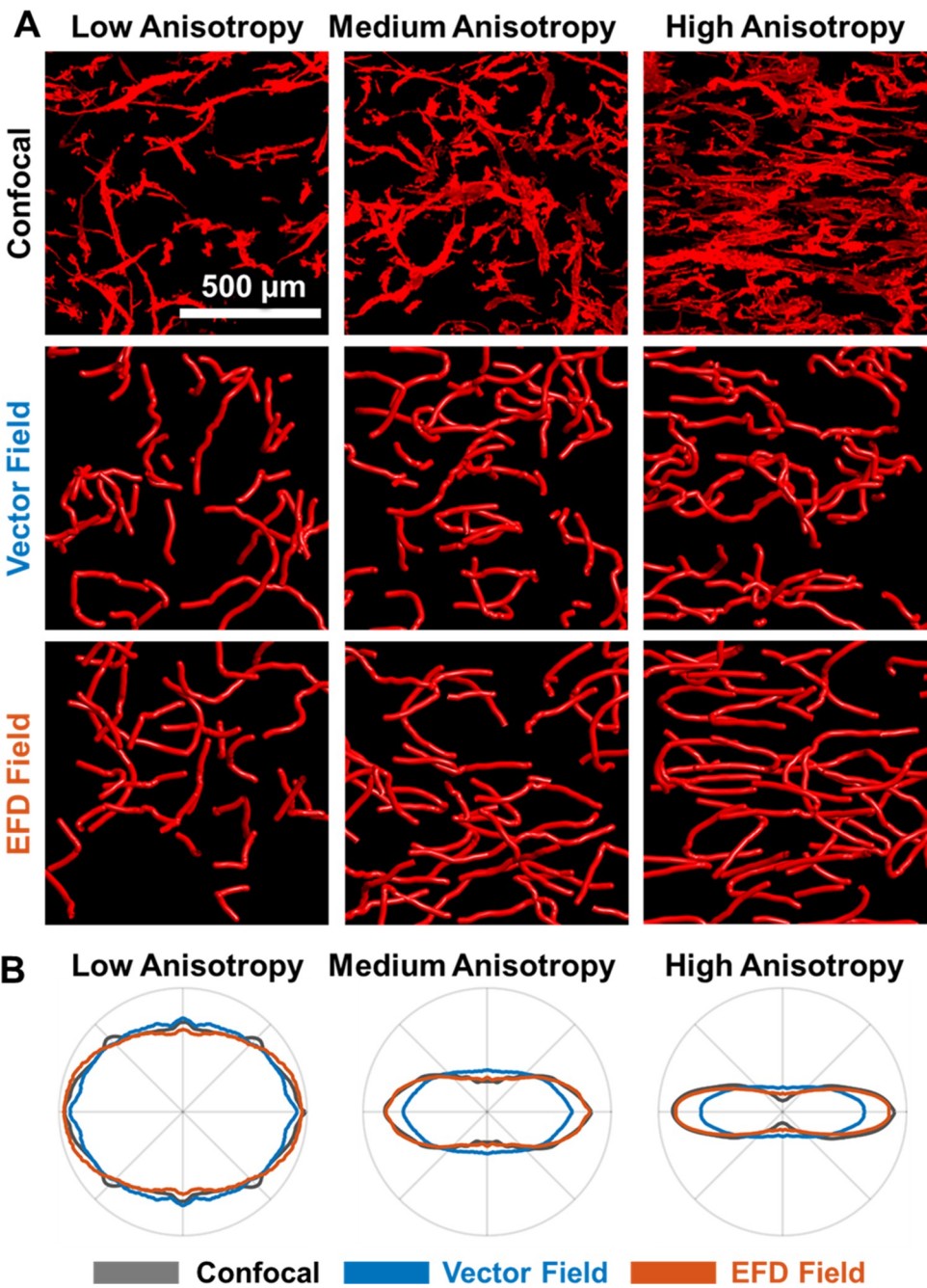

**Fig 3. Comparison of experimental and simulated angiogenesis. A.** Z projections of experimental (confocal) and simulated microvascular networks grown in low, medium, or high anisotropy collagen from our previous study [1]. Qualitative agreement was visible for all cases of low and medium anisotropy but vector field simulations visually differed from experiments and EFD simulations at high anisotropy. Depth of field = 200 μm. **B.** Averaged microvessel ODFs for each experimental or simulated case were projected onto the XY plane to simplify comparison since there was little growth in the Z direction. Microvessel orientations from EFD simulations were in good agreement with experimental microvascular ODFs at all three levels of anisotropy. In contrast, microvascular ODFs from vector field simulations diverged from the experimental data for the cases of medium and high anisotropy.

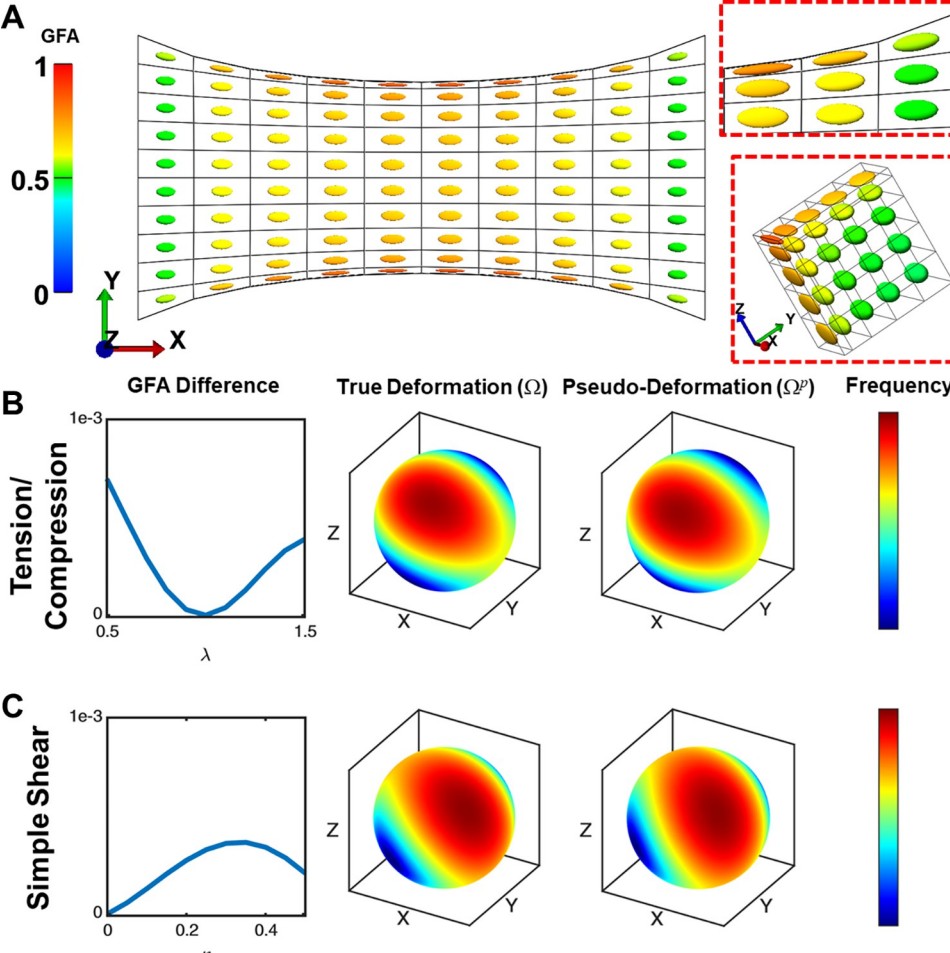

**Fig 4. Visualizations and verification of EFD pseudo-deformation. A.** Visualization of pseudo-deformed collagen fibril ODFs (glyphs colored by GFA) during large-scale uniaxial tension in biphasic materials. Cutouts highlight scaling and rotation at the top-right model corner and along the center row of elements (quarter-symmetry view from the edge to the center). **B-C.** Comparison between ODFs undergoing "true" deformation ($\Omega$) and ODFs undergoing pseudo-deformation ($\Omega^p$) for tension/compression (stretch ratio $\lambda \in [0.5, 1.5]$) and simple shear (shear ratio $\kappa \in [0, 0.5]$) of a single element. Differences in GFA were less than 1e-3 for all cases, which indicated good agreement between $\Omega$ and $\Omega^p$. Heat maps of the 3D ODFs were generated for the test cases with the highest strain to demonstrate the agreement in ODF magnitude and orientation between $\Omega$ and $\Omega^p$.

region of the domain then three cases were developed: 1) baseline case—all regions were iso-tropic (Fig 5A, glyphs), 2) positive gradient case—horizontal anisotropy increased across the domain (Fig 5B, glyphs), and 3) negative gradient case–anisotropy decreased across the domain (Fig 5C, glyphs).

In the baseline case, microvessels vascularized the middle region but failed to grow into the distal region (Fig 5A–5D, and 5E). In contrast, positive and negative anisotropy gradients guided microvessels to the distal end of the simulation domain (Fig 5B–5E; 1 Way ANOVA, Sidakholm post hoc, F [2,29] = 170.12, $p<0.001$). Interestingly, long-range vascularization of the distal region did not differ between positive and negative anisotropy gradients (Fig 5E; post hoc, $p = 0.74$). Short-range vascularization of the middle region also significantly increased due to anisotropy gradients (Fig 5D; 1 Way ANOVA, Sidakholm post hoc, F [2,27] = 426.05, $p<0.001$). Notably, the negative anisotropy gradient case had a significantly greater

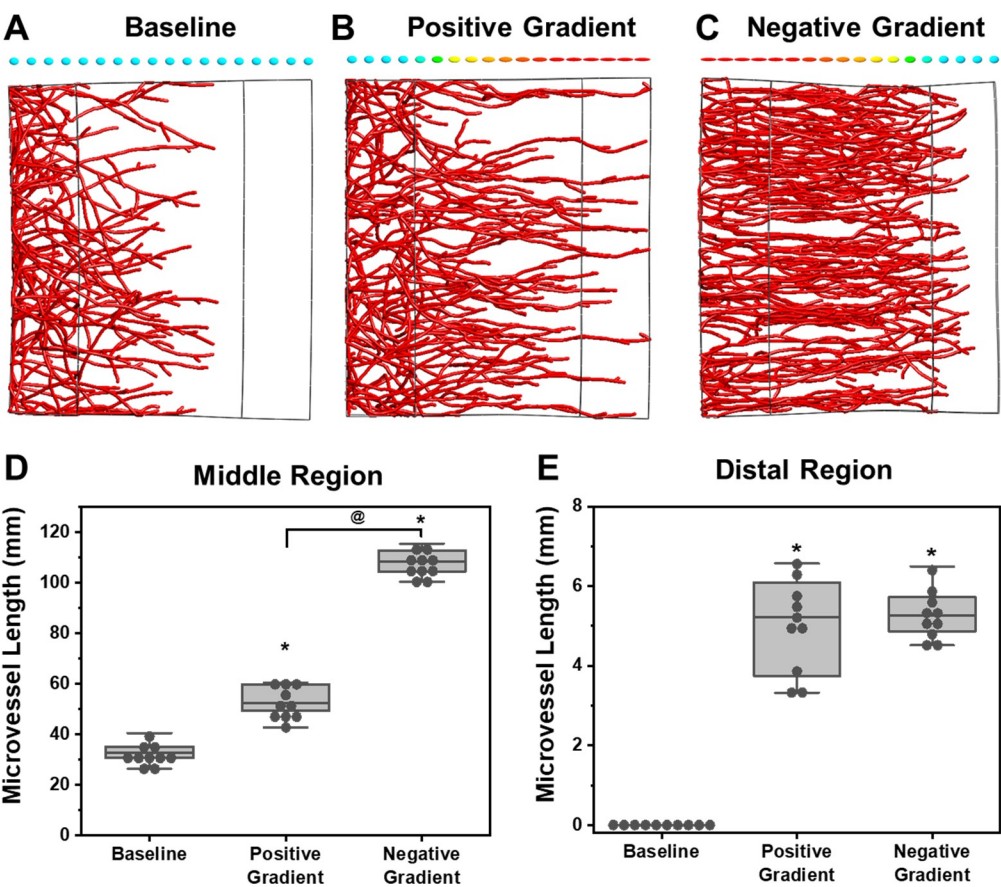

**Fig 5. Predictions of microvascular growth in response to spatial anisotropy gradients.** Simulations were performed with microvessels seeded on the proximal (left) end of a rectilinear domain. Growth was simulated across an isotropic matrix (baseline) or a matrix characterized by a positive or negative anisotropy gradient. Local matrix alignment is indicated by ellipsoidal glyphs above each representative image with the color indicating the anisotropy (blue: low; red: high). Microvessels in the baseline model failed to reach the distal region after 12 days. Anisotropy gradients resulted in increased vascularization of the middle and distal regions. A negative anisotropy gradient resulted in the most vascularization in the middle region, although there was no difference in vascularization of the distal region between gradient cases (1 way ANOVA with Sidakholm post hoc. *: $p < 0.05$ w.r.t baseline; @: $p < 0.05$ post hoc pairwise comparison).

impact on vascularization of the middle region than positive gradients as well (Fig 5B–5D, post hoc, $p < 0.001$).

We also performed these simulations using the vector field method to compare predictions with those from EFD simulations (Supporting Methods F in S1 Text and S9A–S9C Fig). Vessels failed to reach the distal region over a period of 12 days for all vector field simulations (S9D–S9F Fig; 1 Way ANOVA, Sidakholm post hoc, F $[2,27]$ = 62.07, $p < 0.001$). Further, only a negative anisotropy gradient was predicted to enhance vascularization of the middle region (S9G Fig; post hoc, $p < 0.001$).

## Tumor-associated structural interfaces passively recruit microvessels

Tumor-associated collagen signatures (TACS) are unique, heterogeneous configurations of matrix density and anisotropy found in the stroma/desmoplasia near some solid tumors [13, 31, 36]. Certain TACS are associated with an enhanced ability to recruit cells from the stroma, resulting in aggressive cancers and poor prognoses. Thus, we used AngioFE to explore how 5

different TACS affect tumor vascular recruitment during cancer tumorigenesis. Similar to before, a rectilinear simulation domain was generated with three regions: 1) the peritumoral stroma containing the parent microvessels, 2) a thin interface, and 3) a tumor (Fig 6). The density and anisotropy of the interface was modified to reflect clinically observed TACS.

We discovered a significant relationship between TACS presentations and the degree of tumor vascularization (1 Way ANOVA, Sidakholm post hoc; F [5,54] = 52.05, $p<0.001$). In the baseline case (no difference in structure between regions), microvessels grew across the interface from the periphery into the tumor (Fig 6). A high-density interface (TACS-1) greatly reduced tumor vascularization (post hoc, $p<0.001$). Fibril alignment oriented along the interface (TACS-2) led to microvascular growth around the interface, but there was no change in tumor vascularization when compared to the baseline case (post hoc, $p = 0.1$). In contrast, fibril alignment across the interface (TACS-3) facilitated the greatest tumor vascularization overall (post hoc, $p<0.001$). The combined presence of elevated density and fibril alignment along the interface (TACS-1+2) resulted in tumor vascularization comparable to TACS-1 (post hoc, $p = 0.39$). However, in this case, the vessels continued to grow along the interface rather than stop in their tracks as was seen with TACS-1. Finally, the combined presence of elevated density and fibril alignment perpendicular to the interface (TACS-1+3) resulted in a similar degree of tumor vascularization as the baseline case (post hoc, $p = 0.044$, testwise $\alpha = 0.0127$).

We repeated these simulations using a vector field approach (Supporting Methods F in S1 Text). Trends in tumor vascularization were similar aside from the TACS-1+3 case. The vector field simulations predicted that TACS-1+3 would result in less tumor vascularization than the baseline case (S10 Fig).

## Discussion

We developed a new approach to predict the guidance of neovessels during angiogenesis that accounts for the simultaneous influence of the ECM collagen orientation, degree of anisotropy, and density. Cells dynamically interact with the matrix during migration; microvessel sprouts constantly probe the matrix in time-series images of *in vitro* angiogenesis [37]. In our simulations, vessel tips grown using the EFD field approach continuously sampled the local collagen orientations, emulating filopodia probing during angiogenesis. As a result, these simulations accurately predicted anisotropic guidance in agreement with prior experimental measures. In contrast, vector field simulations under-predicted anisotropic guidance (e.g., less vessels were oriented along the X direction in Fig 3). There are some reasons why these differences emerged. First, vector fields were only generated during model initialization (i.e., the collagen orientation is determined by sampling the ODF once). Thus, individual vectors could be randomly assigned orientations that differed greatly from the primary collagen direction. Furthermore, isotropic and near-isotropic ODFs are incompatible with vector fields, which distilled a 3D distribution into a single orientation. In contrast, EFDs were sampled at the beginning of each growth step, which better captured the underlying distribution and was compatible with isotropic ODFs. The final disadvantage of the vector field approach is that neovessels growing through the same finite element will be guided by the same vectors, resulting in similar trajectories despite isotropic underlying collagen networks. In contrast, EFDs allowed neovessels to sample unique directions from the same local EFD, resulting in different trajectories.

We theorized that mesh refinement could improve the ability of vessels in the vector field simulations to "probe" their local environment; however, this was not the case as mesh refinement had little effect on vessel guidance. Thus, differences between approaches likely occurred due to their differing interpolation methods. The vector field approach interpolated the collagen fibril direction from the finite element nodes via linear weighted averaging of vectors. This

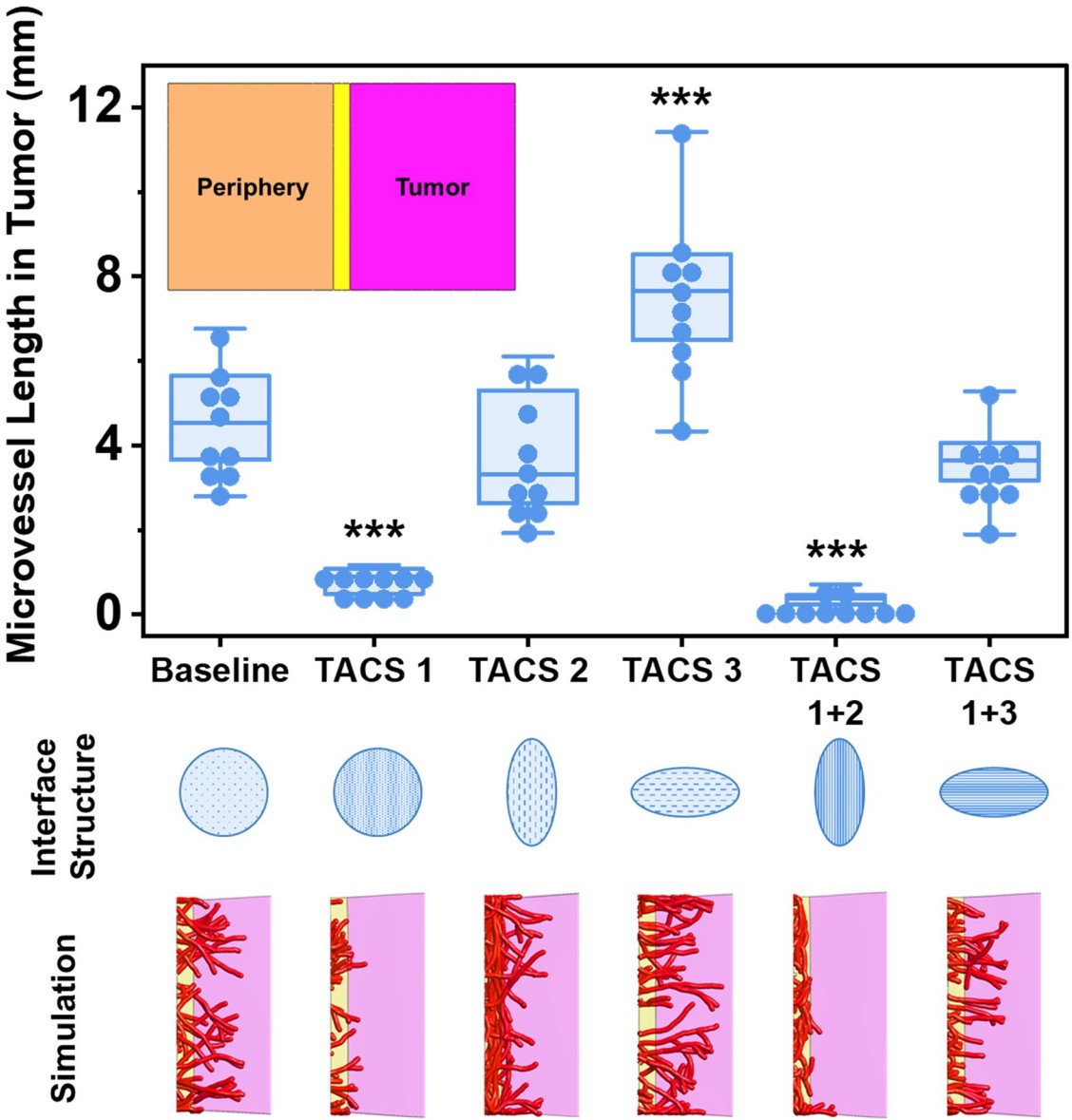

**Fig 6. Tumor associated collagen signatures (TACS) differentially facilitate neovessel recruitment.** Microvessels were simulated to originate in the stroma of a tumor periphery (inset, orange). Microvessels grew within the periphery toward the tumor (inset, magenta), separated by a structural interface (inset, yellow). Alignment and density of the interface was varied to mimic various TACS. The interface comprised an isotropic collagen ODF (circle), or aligned ODFs (ellipses) running perpendicular or along interface. The interface density was either low or high. Representative Z projections of the interface and tumor region are presented at the bottom. High interface density reduced the length of microvessels that crossed into the field (TACS-1 interfaces). Interface alignment along the tumor (TACS-2) deflected vessels or trapped them within the interface, while alignment radiating from the tumor facilitated vascular invasion (TACS-3). Fibril alignment in TACS-3 nullified the effects of increased matrix density. 1 way ANOVA. ***: $p < 0.001$ w.r.t. baseline.

method failed to incorporate the properties of the underlying ODF [33]. In contrast, EFD interpolation was accurate because it calculated a geodesic average, which preserved the properties of the underlying orientation distributions. We can extend this reasoning to the observation that discrete vector field simulations displayed higher sensitivity to elevated sprout tractions than EFD simulations. As cellular tractions intensified, nodal vectors rapidly reoriented towards tractions, which skewed the arithmetically averaged fibril orientation. Thus,

additional care must be taken when studying pathologies characterized by elevated cell tractions (due to biochemical signaling, pharmaceuticals, mutations, etc.), or when the initial vascular density (cell concentration) is much greater.

The degree of anisotropy has not been widely used as a model input in studies of cell migration, despite evidence that the degree of anisotropy affects cell behaviors and guidance [1, 2, 5, 8, 20, 25, 34]. Some contributions to the literature have implicitly modeled this phenomenon, such as 2D simulations of angiogenesis by Sun et al. They incorporated a "matrix conductivity tensor" as an analogue for anisotropy to study how heterogeneously oriented matrices affect vascular growth [24]. Enhanced vessel growth was observed along directions of higher conductivity. However, they concluded that matrix anisotropy decreased vessel growth and increased network tortuosity, contradicting new experimental evidence. This discrepancy can be explained by three factors. First, anisotropy was modeled at a different scale and there was only local alignment (i.e., anisotropy was high but each ODF's orientation was globally random). Their mesh resolution was about the width of a microvessel, at which point the parameters of a continuous fibril distribution becomes sensitive to individual fibril bundles. Additionally, vessels respond to long-range matrix stiffness and anisotropy gradients, which could not be identified by an over-refined mesh [3, 38]. Second, vessel proliferation/extension rates were insensitive to the degree of anisotropy. Third, vascular persistence was not modeled. This last factor piqued our interest, as we had similarly noticed vessel tortuosity when persistence was diminished in our sensitivity studies (Supporting Methods C in S1 Text and S14 Fig). In this case, both modeling frameworks support the theory that pathological vessel tortuosity emerges due to upregulated mechanosensitivity and downregulated persistence [39].

The framework presented here extends methods to simulate anisotropic guidance of cell continua to discretely modeled cells. First, we modified the pseudo-deformation approach of Barocas et al., which approximated EFD deformations as the ellipsoid associated with **B**, the left Cauchy-Green deformation tensor [11, 40]. They appropriately identified the orientation of the deformed EFD but scaled the axial lengths by the square of the eigenvalues of the left stretch tensor **V**. Although this was inconsequential for prior applications (which only used the orientation of the pseudo-deformed EFD as a model input), it was necessary for us to make modifications since our growth model relied on the degree of anisotropy, which depends on physical levels of stretch. We verified the accuracy of our modified pseudo-deformation in the context of various non-affine deformations. The second novelty of this study is the use of stochastic methods for discrete cells to "probe" the matrix by sampling 3D ODFs. In prior modeling approaches, cell migration primarily followed the collagen principal direction and stochastic variation was introduced via random or Brownian processes [20, 25, 41]. Here, we sampled EFDs, which introduced a stochastic component that was directly related to the underlying matrix structure. Further, we avoided tedious calculations of ellipsoidal integrals by using Monte-Carlo methods to sample EFDs [11].

We simulated microvascular growth along a positive or negative gradient of matrix anisotropy to determine how different spatial configurations of matrix anisotropy are involved in neovessel guidance. Short-range and long-range vascularization of the middle and distal regions were increased by positive and negative anisotropy gradients, suggesting that anisotropy gradients can passively recruit neovessels across various length scales. Unexpectedly, vascularization of the distal region was comparable between the positive and negative gradient cases. However, vascularization of the middle region was greater for the negative anisotropy gradient. In this scenario, positive gradients were more efficient at vascularizing the far region. These differences emerged due to the dependence of the vessel elongation rate on the local anisotropy. Collectively, these results highlight the ability of anisotropy gradients passively recruited microvessels. Anisotropy gradients are found in both healthy and diseased tissues.

For instance, connective tissues prominently feature anisotropy gradients that provide mechanical function and are distributed at the level of individual tissues (e.g., skin, long bones, articular cartilage, etc.) as well as boundaries between adjacent tissues (e.g., attachments to fascia, joints, synovia, entheses, etc.) [42–47]. Wounds may disrupt the native tissue anisotropy (e.g., tendon & ligament), and scars may amplify tissue alignment [48, 49]. Deviations in collagen organization also arise from disease (chronic inflammation, arthritis) and old age. In extreme cases, microdamage accumulation can erode tissue boundaries, allowing vessels to invade traditionally avascular niches and induce inflammation [47, 50–53].

Previously, we demonstrated that the proangiogenic effects of matrix anisotropy are attenuated by elevated matrix density *in vitro* [1]. Changes in both matrix anisotropy and density are common at structural interfaces between adjacent tissues. For instance, a number of cancers are characterized by elevated density and/or fibril alignment in the region surrounding the tumor. These hallmarks were termed tumor-associated collagen signatures (TACS) by Provenzano et al. [13]. To test how various TACSs affect tumor vascularization, we simulated angiogenesis near the tumor-stroma interface. The density and anisotropy of the interface were modified to resemble either a single TACS or a clinically relevant combination of multiple TACSs [13, 36, 54]. TACS-1 is characterized by dense collagen around the tumor periphery that forms due to hypertrophic cell growth and myofibroblast collagen deposition [55]. As expected, this type of interface prevented most neovessels from vascularizing the tumor in our simulations. TACS-2 presents as fibril alignment along the surface of the tumor, which can form as the tumor expands and stretches collagen fibrils surrounding it [56]. In our simulations, vessels grew into this interface and began to reorient along it, but still invaded and vascularized the tumor to a similar degree as the baseline case. The combined TACS-1+2 resulted in the least tumor vascularization. Our findings mostly agree with prior simulations of angiogenesis based on an *in vitro* model of angiogenesis across tissue barriers [4]. The present simulations differ in that the interface was more vascularized by TACS-2 and TACS-1+2 because vessel growth increased with matrix anisotropy. These vessels would be susceptible to recruitment by tumor cells, suggesting that TACS-2 increases the odds of vascular invasion by recruiting vessels to the tumor interface. We also investigated the effects of TACS-3, which is characterized by spiculated collagen fibrils radiating outward from the tumor and poor prognoses [57]. TACS-3 is presumed to emerge due to outward cell migration from the tumor or intense radial contractions by the tumor [13, 56]. Our simulations predicted similar outcomes associated with TACS-3 prognoses, as evident by the large increase tumor vascularization. Further, a TACS-1+3 interface was predicted to facilitate baseline levels of tumor vascularization since anisotropy emboldened vessels to persist in spite of tissue density. In summary, we demonstrated that clinical relationships between tissue structure and solid tumorigenesis could be predicted by simultaneously accounting for matrix density, orientation, and the degree of anisotropy. Our findings support the growing interest in stromal-based cancer diagnostics and therapeutics [58]. In the future, we will expand AngioFE to include biochemical signaling to explore the coordination of biophysical and biochemical stimuli during microvessel guidance across tissue structures.

In our predictive simulations, microvessel velocity increased with matrix anisotropy and decreased with matrix density based on experimental observations [1]. This approach can be extended to simulate the roles of other anisotropic biophysical and biochemical stimuli. For instance, there is evidence that microvessels grow along the primary direction of stretch (1$^{st}$ principal strain) and avoid compression (3$^{rd}$ principal strain) [3]. Additionally, the mechanical model can accommodate anisotropic fibril stiffness that emerges from differing fibril properties (as modeled by $\xi$ which varies with fibril diameter, crosslink types, etc.) or variations in fibril stretch along various orientations (e.g., fibrils assume a kinked conformation when

relaxed and a linear conformation when engaged in tension). Thus, the presented approach could easily be adapted to evaluate hypotheses of mechanoregulation based on tissue deformation. Similarly, diffusion and permeability tensors can be used to evaluate the role of anisotropic diffusion and matrix porosity on cell growth and guidance.

## Methods

### Ethics statement

The animal study was reviewed and approved by University of Utah Institutional Animal Care and Use Committee (IACUC).

### In-vitro pre-alignment of collagen hydrogels and microvessel culture

Experimental data for this study were derived from previously published studies in which microvessels were cultured at 3 levels of ECM anisotropy (low, medium, and high anisotropy) as well as 2 levels of ECM density (low or high) [1]. The methods used to characterize ECM architecture and the morphology of microvascular networks after 10 days of culture are summarized below.

Type-I collagen hydrogels were cast at low (3 mg/mL) or high (4 mg/mL) density in rectangular chambers flanked with steel mesh anchors at the ends before incubation at 37˚C (rat-tail tendon, Corning Inc., Corning, NY). Steel mesh anchors were then stretched between 0 and 20% along the long axis at 25 minutes after casting in order to pre-align the ECM collagen fibrils. Gels were cut free from mesh anchors after 1 day then transferred to phosphate buffered saline or cell-culture media depending on the experiment. The ECM collagen microstructure was visualized via second harmonic generation (SHG) imaging (Fig 2). SHG images were then processed and orientation distribution functions (ODFs) were extracted via fast Fourier transforms using a custom MATLAB script (MathWorks, Natick, MA). Ellipsoids were fit to the ODFs, which allowed the calculation of the anisotropy and ellipsoid semiprincipal axes associated with each level of collagen fibril alignment and density.

Intact microvessel fragments were isolated from male Sprague Dawley retired breeder rats using an established protocol [59]. Microvessel fragments were suspended in liquid collagen at low or high density before casting gels in rectangular chambers. Vascularized gels were pre-strained to achieve low, moderate, or high anisotropy, resulting in 6 unique combinations of matrix anisotropy and density. These gels were similarly cut free from the steel mesh anchors after 1 day. Gels were fixed with 2% paraformaldehyde (VWR, Radnor, PA) after 10 days of culture, stained for endothelial cells (Isolectin; ThermoFisher), and then imaged via confocal microscopy. Confocal volumes were processed in AMIRA (ThermoFisher, Waltham, MA) which allowed the extraction of microvessel network ODFs and vascularity (mm vessels / mm$^3$ image volume). A set of vascularized gels were also fixed, stained, and imaged on Day 1 to provide baseline measures of vascularity and vessel polarization that would later be used to generate the initial simulation conditions.

### Pseudo-deformation and interpolation of ellipsoidal fibril distribution structure tensors

We assumed that EFDs approximate collagen ODFs for spatial scales on the order of 100 μm based on prior collagen structural imaging [3, 20]. Further, we assumed the individual fibrils in an ODF deform independent of each other (i.e., no interaction during deformation). These assumptions allowed us to approximate 3D collagen distributions with 6-term symmetric positive-definite tensors (SPDs) which are inherently ellipsoidal. In general, a fiber distribution

that is ellipsoidal in the reference (unloaded) configuration of a material does not remain ellipsoidal upon finite deformation, since the semi-principal axes of the ellipsoid may not remain orthogonal. In order to maintain the symmetry properties of the EFDs after deformation, we assumed that EFDs remain ellipsoidal but experience affine deformation (finite rotation and stretching) of their principal radii of curvature. This was necessary to conveniently update and sample EFDs, as well as to spatially interpolate local matrix ODFs and visualize the deformed ODFs.

The collagen fibril matrix undergoes deformation due to cellular tractions, boundary constraints, external forces, and fluid flow [11, 35]. Under the action of material deformations an SPD does not necessarily map to an SPD; i.e., non-affine mappings will deform initial EFDs $\Omega^0$ into ODFs $\Omega$ which may lack symmetry and/or orthogonality (Fig 7A). In general, deformed collagen ODFs are ellipsoidal when observing neighborhoods at physical scales of ~100 μm. Thus, we approximated a symmetric, orthogonal ODF $\Omega^p$ that approximated the stretch, orientation, and anisotropy of $\Omega$. This was accomplished by modifying

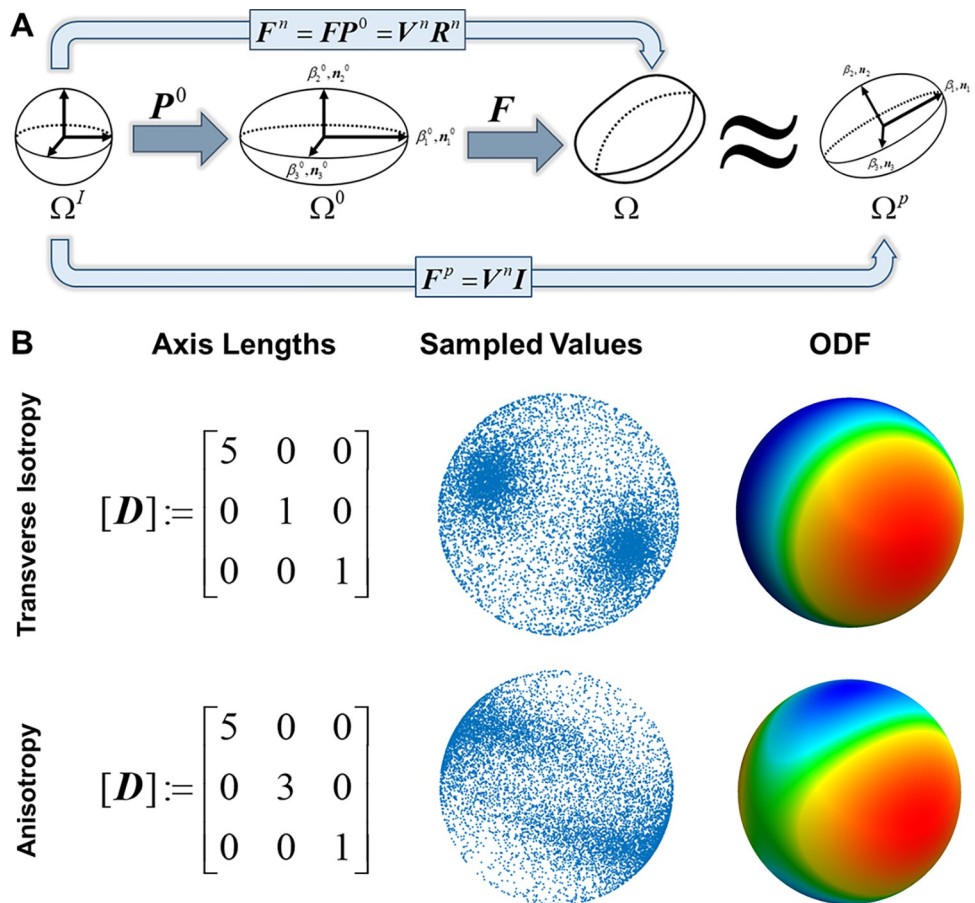

**Fig 7. Pseudo-deformation concept map & Monte-Carlo EFD sampling visualization. A.** EFD deformation & pseudo-deformation concept map. Vectors originating from the center of a unit sphere $\Omega^I$ were mapped by $\boldsymbol{P}^0$ to the undeformed EFD $\Omega^0$. After deformation by $\boldsymbol{F}$, the fibrils of $\Omega^0$ were stretched and rotated, yielding the deformed ODF $\Omega$. The net mapping from $\Omega^I$ to $\Omega$ was given by $\boldsymbol{F}^n$. Polar decomposition of $\boldsymbol{F}^n$ yielded $\boldsymbol{F}^n = \boldsymbol{V}^n \boldsymbol{R}^n$. An EFD $\Omega^p$ that closely approximated $\Omega$ was found by applying the pseudo-deformation $\boldsymbol{F}^p = \boldsymbol{V}^n \boldsymbol{I}$ to $\Omega^I$. **B.** Visualizations of EFD Monte-Carlo sampling are provided for the cases of transverse isotropy (semiprincipal axis lengths $\beta_1 = 5$, $\beta_2 = \beta_3 = 1$) and orthotropy (semiprincipal axis lengths $\beta_1 = 5$, $\beta_2 = 3$, $\beta_3 = 1$). Sampled directions were mapped onto the surface of a unit sphere. The ODF was generated by summing points on the surface of the unit sphere that corresponded to faces on an icosahedron.

the methods previously outlined by Barocas et al. [11, 40]. First, we considered the SPD tensor $\boldsymbol{P}^0$, which describes the initial collagen EFD. The components of $\boldsymbol{P}^0$ are given by the semiprincipal axis magnitudes $\beta_i$ and orientations $\boldsymbol{n}_i^0$ which are related via the spectral decomposition

$$\boldsymbol{P}^0 = \sum_{i=1}^{3} \beta_i \boldsymbol{n}_i^0 \otimes \boldsymbol{n}_i^0 \tag{2}$$

Further, $\boldsymbol{P}^0$ can be used to map radii from a unit sphere $\Omega^I$ to radii of the undeformed EFD $\Omega^0$ (Fig 7A). We next mapped the radii of $\Omega^0$ to $\Omega$ via the deformation gradient $\boldsymbol{F} = \partial\boldsymbol{x}/\partial\boldsymbol{x}^0$. The net transformation from $\Omega^I$ to $\Omega$ was given by the mapping

$$\boldsymbol{F}^n = \boldsymbol{F}\boldsymbol{P}^0 = \boldsymbol{V}^n \boldsymbol{R}^n \tag{3}$$

The polar decomposition of Eq [3] gives the SPD $\boldsymbol{V}^n$ and the orthogonal rotational tensor $\boldsymbol{R}^n$. Previously, Barocas et al. [40] approximated the pseudo-deformed EFD via an analog of the Left-Cauchy deformation tensor $\hat{\boldsymbol{B}}$ given by

$$\hat{\boldsymbol{B}} = \boldsymbol{F}\boldsymbol{P}^0 \boldsymbol{P}^{0T} \boldsymbol{F}^T = \boldsymbol{F}^n \boldsymbol{F}^{nT} \tag{4}$$

Here, $\hat{\boldsymbol{B}}$ differs from the Left-Cauchy deformation tensor $\boldsymbol{B}$ since $\hat{\boldsymbol{B}}$ transforms points from a unit sphere to the final state (points $\boldsymbol{x}^i$ on the unit sphere surface to $\boldsymbol{x}$) while $\boldsymbol{B}$ transforms points between the initial and final states (points $\boldsymbol{x}^0$ on the surface of an undeformed EFD to $\boldsymbol{x}$). However, we were concerned with the physical levels of stretch, whereas $\hat{\boldsymbol{B}}$ was related to the squared stretch of $\boldsymbol{V}^n$. Thus, we approximated $\boldsymbol{F}^n$ with the SPD $\boldsymbol{F}^p$ that replaced $\boldsymbol{R}^n$ with the identity $\boldsymbol{I}$ in order to remove non-affine deformations:

$$\boldsymbol{F}^p = \boldsymbol{V}^n \boldsymbol{I} = \boldsymbol{V}^n \tag{5}$$

$\boldsymbol{F}^p$ allowed us to map radii from $\Omega^I$ to the pseudo-deformed EFD $\Omega^p$ that approximated the true $\Omega$ (Fig 7A). The semiprincipal axis magnitudes $\beta_i$ and directions $n_i$ of $\Omega^p$ were calculated from the spectral decomposition of $\boldsymbol{F}^p$ as previously done in Eq [2].

In a finite element analysis, the deformation gradient is evaluated at the element integration points within the finite element domain. However, directed guidance of angiogenesis relies on knowledge of the fiber distribution throughout the element domain. Therefore, EFDs were interpolated across the spatial domain using a geodesic weighted average [33]. The local EFD at a growing vessel tip was interpolated from EFDs stored at the integration points of the element containing the tip where weights of each EFD were determined by the finite element shape functions.

The fractional anisotropy of an EFD was calculated from its principal radii of curvature as

$$\text{FA} = \sqrt{\frac{1}{2}\frac{(\beta_1 - \beta_2)^2 + (\beta_2 - \beta_3)^2 + (\beta_3 - \beta_1)^2}{\beta_1^2 + \beta_2^2 + \beta_3^2}} = \frac{\text{std}(\beta_i)}{\text{rms}(\beta_i)} \tag{6}$$

where std is the standard deviation of the ellipsoid radii $\beta_i$ and rms is the root mean square [60]. Importantly, the fractional anisotropy is only applicable to SPDs since the metric assumes an underlying ellipsoidal distribution.

## Validation of pseudo-deformed EFDs against true orientation distribution function deformation

EFDs were subjected to either uniaxial tension/compression with applied stretch ratios $\lambda \in$ [0.5, 1.5] or shear with applied shear ratios of $\kappa \in [0, 0.5]$ (Supporting Methods E in S1 Text and S5–S8 Figs). The initial EFDs, $\Omega^0$, were specified as isotropic, transversely isotropic (1 preferred direction), or planarly anisotropic (2 orthogonal preferred directions of equal magnitude) to evaluate how pseudo-deformation accuracy depended on the underlying fibril distribution. Each point, $s^0$, of an icosahedron mesh containing 10,242 nodes was mapped to the surface of either the true deformed ODF $\Omega$ (derived from $F^n$), or the pseudo-deformed ODF $\Omega^p$ (derived from $F^p$) using a custom MATLAB script [61]. Deformed positions were mapped back to the surface of a unit sphere so that they could be assigned to faces of an icosahedron. Points $s$ on the surface of $\Omega$ were calculated and mapped to the unit sphere via:

$$s = \frac{F^n \cdot s^0}{\|F^n \cdot s^0\|} \quad [7]$$

Here, the values of the probability density function of $\Omega$ were computed by determining the fraction of points $s$ that mapped onto each triangular facet of an icosahedron. Similarly, the values of the probability density function of $\Omega^p$ were calculated by mapping points $s^p$ on the surface of $\Omega^p$ to the unit sphere icosahedron via:

$$s^p = \frac{F^p \cdot s^0}{\|F^p \cdot s^0\|} \quad [8]$$

Two quantitative methods were used to assess similarity between deformed and pseudo-deformed ODFs. First, the generalized fractional anisotropy (GFA) for the true and pseudo-deformed distributions via:

$$\text{GFA} := \frac{\text{std}(\Omega(r_i))}{\text{rms}(\Omega(r_i))} \quad [9]$$

Notably, Eq [9] simplifies to Eq [6] for ellipsoidal distributions. The second quantitative method involved calculation of the Fisher-Rao distance, a Riemannian metric quantifying the difference in probability distributions between the pseudo-deformed EFD and the true deformation [62, 63]. This metric ranges between 0 and 90˚, where 0˚ indicates a complete overlap in distributions, and 90˚ indicates no overlap between distributions. The MATLAB scripts used for this analysis are available at github.com/febiosoftware/AngioFE under the documentation section.

## Random sampling of EFDs

The direction that a vessel tip grew in AngioFE depended on the prior direction $\psi$ and the predominant local collagen fibril orientation $\theta$. Previously, the collagen fibril direction was represented as a field of vectors stored at the finite element nodes. However, vessels *in vitro* and *in vivo* probe fibrils oriented in many different directions and ultimately decide to grow along a single one of those directions. To simulate this stochastic phenomenon, we introduced a method that randomly samples a single fibril orientation from the local collagen EFD. In our new approach, $\theta$ was sampled from the collagen EFDs using a Monte-Carlo approach. For an EFD with an arbitrary basis (i.e., 3D rotation) this was performed by sampling a direction in the global basis before rotating it into the EFD basis. First, the EFD was split via Eigen-decomposition to yield $\beta_i$, the semiprincipal axes in the global basis. Random direction vectors $r$ were

generated inside of a cube bounding the EFD such that each component of $r$ was constrained by $r_i \in [-\max(\beta_i), \max(\beta_i)]$. Points were accepted if they fell within the ellipsoid as determined by the inequality:

$$\frac{r_1^2}{\beta_1^2} + \frac{r_2^2}{\beta_2^2} + \frac{r_3^2}{\beta_3^2} < 1 \qquad [10]$$

Visualizations of the 3D sampling method for transverse isotropy and anisotropy are presented in Fig 7B.

## Interpolation of fibril directions and EFDs

Vessel tips can occupy any position within the simulation domain, but data about the local fibril distributions are stored at the local finite element integration points. Thus, we developed a method to interpolate EFDs from the integration points to the position of a vessel tip. Simulations were performed using either a vector-field or an EFD approach for representing collagen fibril ODFs. Vector fields were spatially interpolated via a linear weighted sum of the orientations stored at each integration point in the finite element mesh [20]. The weights were calculated using the finite element shape functions based on the current position of the vessel tip in the element. EFDs were defined at the finite element integration points as SPDs. The local EFD was calculated via a geodesic weighted sum of the integration points to ensure interpolated tensors maintained orthogonality and positive-definiteness [33]. This was accomplished using algorithms 1, 2, and 5 from [33].

## FEBio + AngioFE computational model of microvascular growth

AngioFE is developed as a plugin for FEBio to simulate microvascular growth and cell-matrix interactions within the FEBio framework [21]. Briefly, microvessel parent fragments are represented as line segments seeded within a finite element mesh. The end of each line is a vessel tip that is capable of growing and contracting the surrounding matrix. Textual and graphic summaries of the FEBio + AngioFE modeling framework are presented in the results ("*AngioFE model formulation*", Fig 1). AngioFE version 3.0 and FEBio 3.7 were used for all simulations. AngioFE version 3.0 was developed to address prior limitations in the growth model and to introduce new mechanisms of directional guidance by ECM fibril orientation (Supporting Methods H-K in S1 Text). This software is freely available at github.com/febiosoftware/ AngioFE. The builds of FEBio and AngioFE that were used as well as all input files are available in the documentation section of the AngioFE GitHub page.

**Model generation & inputs.** Model geometries, materials, and boundary conditions were prescribed in FEBioStudio. Linear hexahedral elements (Hex8) were used for all simulations. Material behavior in the finite element models was represented with a biphasic (i.e., fluid-solid mixture) viscoelastic constitutive model comprising a mix of a neo-Hookean ground matrix, a continuous collagen fiber distribution, and suspended neo-Hookean neovessels [19, 64]. The moduli of the collagen fibrils varied with strain due to stretch and rotation. Collagen mechanical parameters were derived from prior stress-relaxation experiments of collagen gels [65]. All exposed edges on models were prescribed free-draining boundaries to allow fluid exudation.

**Model initialization.** Model initialization comprised of generating the initial parent microvessels as well as maps of material properties relevant to angiogenesis (ODFs, density). To generate the initial parent microvessels, we first examined confocal images of 1-day-old collagen gel cultures of microvessel fragments prior to sprouting. Probability distributions of the initial vessel lengths and XY plane orientations were extracted from images. Vessels were randomly added to the simulation domain in a manner that recapitulated the image-derived

distributions (Supporting Methods H in S1 Text and S12 Fig). Finally, data maps were generated to project the density and collagen ODF to the finite element integration points.

**Vessel orientation.** The new direction each vessel tip grew, $\boldsymbol{\psi}_{new}$, depended on persistence along the current orientation of the vessel, $\boldsymbol{\psi}$, and contact guidance along $\boldsymbol{\theta}$ [2, 4]. The contact guidance direction was determined by 1) interpolating the local EFD from the integration points, 2) pseudo-deforming the local EFD, and 3) Monte-Carlo sampling of the pseudo-deformed EFD [33]. The direction that each vessel grew, $\boldsymbol{\psi}_{new}$, was determined by the equation

$$\boldsymbol{\psi}_{new} = \boldsymbol{R}(\boldsymbol{\theta}, \boldsymbol{\psi}, \alpha)\boldsymbol{\psi}. \tag{11}$$

The function $\boldsymbol{R}(\boldsymbol{\theta}, \boldsymbol{\psi}, \alpha)$ was formulated as a rotation matrix about the shared orthogonal axis of $\boldsymbol{\psi}$ and $\boldsymbol{\theta}$ (Supporting Methods A in S1 Text and S1 Fig). The fibril weight parameter $\alpha \in [0, 1]$ scaled the rotation from $\boldsymbol{\psi}$ towards $\boldsymbol{\theta}$, such that $\alpha = 0 \rightarrow \boldsymbol{\psi}_{new} = \boldsymbol{\psi}$, and $\alpha = 1 \rightarrow \boldsymbol{\psi}_{new} = \boldsymbol{\theta}$.

**Vessel extension rate.** The length each tip grew per time step was determined by scaling a time-dependent sigmoidal growth function. Unless otherwise noted, the time-dependent growth function was modeled using the time-derivative of a sigmoidal curve, with parameters derived from prior experiments (Supporting Methods G in S1 Text) [1, 4, 20]. The time-dependent growth function was scaled by the 3D Lorentzian function $v(\rho, \text{FA})$ (S11 Fig) to introduce a dependence on both the density ($\rho$) and anisotropy (FA):

$$v(\rho, \text{FA}) = \frac{a_v}{\left(1 + \left(\frac{\rho - \rho_{0,v}}{b_v}\right)^2\right)\left(1 + \left(\frac{\text{FA} - \text{FA}_{0,v}}{c_v}\right)^2\right)} + d_v \tag{12}$$

The total length the tip grew was determined by the product of the time-dependent function and the scalar function. New vessel tips were then grown along $\boldsymbol{\psi}_{new}$. Parameters for Eq [12] were identified from prior microvessel cultures in collagen hydrogels of varying density and anisotropy [1]. After culture, the microvessel network lengths were measured and the growth corresponding to each matrix configuration was normalized relative to measurements from unaligned 3 mg/mL collagen matrices. These relative growth values were then used to fit parameters $a_v$, $b_v$, $c_v$, $\rho_{0,v}$ and $\text{FA}_{0,v}$, in the MATLAB curve fitting toolbox. The parameter $d_v$ was added to sustain minimal, non-negative growth.

## Branching and cell tractions

Vessel branches and cell tractions were added after the main vessel growth step. Branches were generated to appear along recently grown vessel segments based on prior implementations in AngioFE (Supporting Methods J in S1 Text). Traction fields were imposed around growing vessel tips that allowed them to deform their surrounding ECM [19, 34]. The traction magnitude was assumed to increase over time as the vascular network grew and sprouts matured. The traction magnitude was also assumed to decrease with increasing matrix density. Additional information is presented in Supporting Methods L-M in S1 Text.

## Simulations and data collection

All simulations were performed 10 times using unique random engine seeds. The random engine generated the initial vessel positions, initial vessel lengths, the length a vessel grew before branching, and random vectors used to sample directions from EFDs. Results from the 10 simulations were averaged. Simulation parameters and justifications are presented in S1 Table.

## Comparison of discrete and continuous fibril distribution approaches

A parametric study was performed on the collagen fibril orientation weight ($\alpha$) to determine appropriate values to simulate microvessel guidance in a range of matrix anisotropies and densities. This study was performed using either discrete fibril vector fields or continuous EFDs to represent collagen ODFs. The results for each approach were compared to prior experiments of microvascular growth in low, moderate, and high anisotropy matrices at low (3.0 mg/mL) or high (4.0 mg/mL) density [1]. Branching was not simulated for this study to simplify analysis. Vectors for the vector field models were stored at finite element integration points and were randomly sampled from the initial EFDs during the initialization step.

First, the initial parent fragments were generated by AngioFE's initialization step. Initial microvessel orientations were prescribed so that the orientations and lengths of fragments matched experimental measurements prior to sprouting [1]. To do this, vessel segments from confocal images acquired on day 0 of culture were projected onto the XY and XZ planes and the ratio of the resulting ODF semiprincipal axes was calculated. We found that microvessel segments were initially isotropic in the XY plane; thus, the semiprincipal axes for the X and Y direction during initialization are set to $\beta_1^P = \beta_2^P = 1.0$. The semiprincipal axis for the Z direction was varied between 0.1–0.9 until the vessel XZ distribution matched the experimental values. The collagen fibril weight during initialization was set to 0.3.

The collagen fibril semiprincipal axes and collagen fibril orientation weight, $\alpha$, were determined in a similar manner as done for the initialization step. Microvascular growth velocity for each condition was prescribed based on prior experimental measures. The EFD semiprincipal axes $\beta_1$ and $\beta_2$ for each simulation are derived from second harmonic generation (SHG) images of collagen gels aligned by stretch during polymerization [1]. The ratio of $\beta_1$ and $\beta_2$ was determined by fitting ellipsoids to the XY ODF in the images. Next, the weight $\alpha$ was varied between 0 and 1. The resulting microvessel network was projected onto the XY planes to calculate ODFs for guidance in the horizontal plane. Values of $\alpha$ were chosen for each condition and approach and then the parameter $\beta_3$ was varied between 0.1–1.0 to approximate the collagen orientation in the vertical direction.

All simulations for this study were performed using a 1 x 1 x 1 mm$^3$ cube. Trilinear hexahedral finite elements were used to discretize the domain with a size of 0.1 x 0.1 x 0.1 mm$^3$ each. Nodes along the X, Y, and Z-axes were fixed in lateral directions to keep the model centered. Visualizations with similar depths of field as confocal Z projections from prior experiments were obtained by using XY plane cuts.

In the model, fibril orientation and growing neovessels partake in a 2-way feedback loop known as dynamic reciprocity [66]. Neovessel orientation is determined by the fibril orientation, and then the fibril orientation is displaced by tractions of growing neovessels. To determine the influence of vessel traction on this feedback system, the above simulations were repeated with 3 values of traction amplitude (Supporting Methods M in S1 Text).

## Mesh sensitivity study

A mesh sensitivity study was performed to determine how mesh size affected directional guidance. Simulations from the parametric study were repeated with 8, 64, 512, 1000, or 8000 total elements, with element side lengths of 500, 250, 125, 100, and 50 μm respectively. We evaluated the effect of mesh refinement on the ratio of the semiprincipal axes of the ellipsoids fit to ODFs of microvessel growth in the XY plane.

## Simulation of anisotropy gradients during healing

We examined how microvessels respond to gradients of matrix anisotropy. The simulation domain contained a middle region that was relatively isotropic for a baseline simulation or

contained a gradient of increasing anisotropy (positive) or decreasing anisotropy (negative) across the simulation domain. The magnitude of the semiprincipal axis in the X direction $\beta_1$ increased linearly across the middle region from 1.3–5.0. The magnitudes in the Y and Z directions were fixed at $\beta_2 = 1.0$ and $\beta_3 = 0.5$ respectively. Microvessels were seeded on the proximal (left) end of the simulation domain. The flanking end geometries were 0.4 x 2.0 x 0.5 mm$^3$ and the center region geometry was 1.0 x 2.0 x 0.5 mm$^3$. All elements were specified as 0.1 x 0.1 x 0.1 mm$^3$ FEBio hex8 elements. Microvessel cultures were simulated for 12 days using a growth velocity rule that supported linear growth after the first few days of culture until the local vessel volume fraction exceeded a threshold $w_{thresh}$ (Supporting Methods K in S1 Text).

## Simulations of neovascular tumor invasion

The second set of predictive simulations examined the role of fibril alignment and density in preventing or facilitating vessel crossing at a structural tissue interface. The properties of the structural interface varied based on tumor associated collagen signatures (TACS) and our prior *in vitro* interface models [4, 13]. We examined how different TACSs affect the ability of microvessels to cross a structural interface between a tumor and its peripheral stroma. The simulation domain was divided into the tumor (1 x 0.8 x 0.6 mm$^3$), the periphery (1 x 0.8 x 0.6 mm$^3$), and an 80 μm thick interface separating them (1 x 0.08 x 0.6 mm$^3$). The interface geometry was based on prior simulations of microvessel crossing at a tissue interface [4]. Elements were specified as 0.1 x 0.1 x 0.1 mm$^3$ FEBio hex8 elements in the stroma and the tumor, and as 0.1 x 0.04 x 0.01 mm$^3$ FEBio hex8 elements in the interface.

Six scenarios were simulated with modifications to the structure of the interface: a baseline simulation (no change in density or anisotropy), TACS-1 (high-density interface), TACS-2 (fibril alignment along interface), TACS-3 (fibril alignment across interface), TACS-1+2 in combination, and TACS-1+3 in combination. The resulting length of microvessels in the tumor after 10 days was quantified to compare how each TACS facilitated microvascular invasion.

## Statistical analysis

Statistical analyses were performed in Origin 2020b (OriginLab, Northampton, MA). One-way analysis of variance (ANOVA) with a Sidakholm means comparison post hoc test was used for assessing statistical significance. Box plot data are centered at the median with edges at 25% and 75% of the data. Whiskers are at 1.5 x IQR. Significance was measured with a family-wise $\alpha$ set to 0.05.

## Supporting information

**S1 Text. Supporting methods including algorithmic details and additional studies.** The supporting information contains additional explanations and details of algorithmic updates to AngioFE. We have also included parameter and sensitivity studies that we used to guide finite element meshing and parameter selection. Finally, a table is provided with the coefficients used for various parameters in our simulations.
(DOCX)

**S1 Table. Model coefficients and justification.** The coefficients used for various parameters in our simulations are presented in this table. Justifications and any additional assumptions are presented as well.
(DOCX)

**S1 Fig. Visualization of Equation S2.** A vessel tip is centered at the origin. The new direction that a vessel grows depends on the previous direction ($\psi$) and the local collagen direction ($\theta$). A coplanar rotation from $\psi$ to $\theta$ occurs about the axis $u = \psi \times \theta$. The new direction $\psi_{new}$ is found by partially rotating $\psi$ towards $\theta$. The fraction of the whole rotation is determined by $\alpha$ $\in [0, 1]$ where $\alpha = 0$ performs no rotation, and $\alpha = 1$ performs the entire rotation.
(TIF)

**S2 Fig. Sensitivity study–Collagen fibril orientation weight** ($\alpha$). Top: Low-density collagen results for simulations where the fibril weight ($\alpha$) was varied from 0.0–1.0. Here, 0 indicates vessel grow along the persistence direction while 1 indicates that vessels grow entirely along the sampled fibril direction. Plotted is the ratio of the major and minor semiprincipal axes for ODFs of microvascular orientation. Discrete fibril approaches and continuous EFD approaches to representing matrix fibril distributions were compared to prior experimental measures (target) for simulations of growth in low, medium, or high anisotropy collagen. Bottom: Similar results were generated from simulations with high-density matrix collagen.
(TIF)

**S3 Fig. Angiogenesis in high-density collagen of varying anisotropy. A**. Z projections of experimental (confocal) and simulated microvascular networks grown in low, medium, or high anisotropy collagen. Depth of field = 200 μm. **B**. XY-plane ODFs for each experimental or simulated case. Overlap is seen between the confocal and EFD field ODFs for all cases. The presented results are for 4 mg/mL collagen.
(TIF)

**S4 Fig. Mesh refinement study.** Directional guidance was generally unaffected by mesh refinement. A 1x1x1 mm3 cube was meshed between 8 and 8,000 FEBio hex8 elements.
(TIF)

**S5 Fig. Verification of pseudo-deformation–uniaxial tension.** Three different initial EFDs were tested with the EFD designated by $P^0$ (Supporting Methods E in S1 Text). $P^0$ is an EFD representing a uniform, uniaxially aligned, or planarly isotropic (transversely anisotropic) EFD. These EFDs are rotated by $R$, a rotation matrix of 45˚ about the X, Y, and Z-axes. The rotation ensures that deformations do not occur along the EFD principal directions (in which cases the pseudo-deformation and "true deformation" are identical during tension and some shear cases). The resulting GFA for each are compared (deformation solid, pseudo-deformation dashed). The Fisher-Rao distance between ODFs was also calculated for each. The resulting ODFs are visualized via heat maps. Heat maps correspond to the highest level of stretch tested.
(TIF)

**S6 Fig. Verification of pseudo-deformation–biaxial tension.** Three different initial EFDs were tested with the EFD designated by $P^0$ (Supporting Methods E in S1 Text). $P^0$ is an EFD representing a uniform, uniaxially aligned, or planarly isotropic (transversely anisotropic) EFD. These EFDs are rotated by $R$, a rotation matrix of 45˚ about the X, Y, and Z-axes. The rotation ensures that deformations do not occur along the EFD principal directions (in which cases the pseudo-deformation and "true deformation" are identical during tension and some shear cases). The resulting GFA for each are compared (deformation solid, pseudo-deformation dashed). The Fisher-Rao distance between ODFs was also calculated for each. The resulting ODFs are visualized via heat maps. Heat maps correspond to the highest level of stretch tested.
(TIF)

**S7 Fig. Verification of pseudo-deformation–simple shear.** Three different initial EFDs were tested with the EFD designated by $P^0$ (Supporting Methods E in S1 Text). $P^0$ is an EFD representing a uniform, uniaxially aligned, or planarly isotropic (transversely anisotropic) EFD. These EFDs are rotated by $R$, a rotation matrix of 45˚ about the X, Y, and Z-axes. The rotation ensures that deformations do not occur along the EFD principal directions (in which cases the pseudo-deformation and "true deformation" are identical during tension and some shear cases). The resulting GFA for each are compared (deformation solid, pseudo-deformation dashed). The Fisher-Rao distance between ODFs was also calculated for each. The resulting ODFs are visualized via heat maps. Heat maps correspond to the highest level of stretch tested. (TIF)

**S8 Fig. Verification of pseudo-deformation–pure shear.** Three different initial EFDs were tested with the EFD designated by $P^0$ (Supporting Methods E in S1 Text). $P^0$ is an EFD representing a uniform, uniaxially aligned, or planarly isotropic (transversely anisotropic) EFD. These EFDs are rotated by $R$, a rotation matrix of 45˚ about the X, Y, and Z-axes. The rotation ensures that deformations do not occur along the EFD principal directions (in which cases the pseudo-deformation and "true deformation" are identical during tension and some shear cases). The resulting GFA for each are compared (deformation solid, pseudo-deformation dashed). The Fisher-Rao distance between ODFs was also calculated for each. The resulting ODFs are visualized via heat maps. Heat maps correspond to the highest level of stretch tested. (TIF)

**S9 Fig. Vector field simulations of anisotropy gradients. A-C.** Top-down view of the initial fibril directions for each element. Directions were randomly sampled from the element's collagen EFD. Local matrix alignment is indicated by ellipsoidal glyphs above each representative image with the color indicating the anisotropy (blue: low; red: high). **D-F.** Microvessels were seeded on the left end of a rectilinear domain. Growth was either simulated in a relatively isotropic matrix (baseline) or a matrix characterized by a positive or negative anisotropy gradient. Microvessels for all models failed to reach the far region on the right. EFD simulations, on the other hand, predicted vascularization of the far region for both gradient models. **G.** Differences were also observed for growth in the middle region. Here, only a negative gradient resulted in increased vascularization, differing from the EFD simulations where both gradients resulted in increased growth in the middle region. (*: $p < 0.001$ w.r.t baseline; @: $p < 0.001$ pairwise comparison, 1 way ANOVA with Sidakholm post hoc). (TIF)

**S10 Fig. Vector field simulations of TACS and neovessel invasion.** Microvessels were simulated to originate in a tumor periphery (inset, orange). Microvessels grow towards the tumor (inset, magenta) but must cross a tissue interface (inset, yellow). Alignment and density of the interface is varied to mimic various tumor associated collagen signatures (TACS). The interface structure contained either an isotropic fibril ODF (circle) or aligned ODFs (ellipses) running either perpendicular to the tumor or towards the tumor. The interface density was either 3 mg/mL or 5 mg/mL. Representative Z projections of the interface and tumor region are presented at the bottom. High interface density generally reduced the length of microvessels that crossed into the tumor. Fibril alignment towards the tumor facilitated crossing while alignment perpendicular to the tumor deflected vessels or trapped them within the interface. Unlike the EFD simulations, fibril alignment towards the tumor did not nullify the effects of increased matrix density for TACS-1+3. ***: $p < 0.001$. 1 Way ANOVA with Sidakholm post hoc test. (TIF)

**S11 Fig. Visualization of $v(\rho,$FA) (Eq 12).** Eq [12] is a function designed to scale vessel growth based on matrix density and anisotropy. Scaling decreases velocity with increased matrix density ($\rho$) and increases with anisotropy (FA). Left: contour level plot with 0.2 level increments. Right: 3D contour plot with experimental points overlaid.
(TIF)

**S12 Fig. Sampling of initial vessel lengths based on confocal images.** Top: The initial length of each discrete microvessel was sampled from a rational function P($L_0$) that had been fit to experimental measures of initial microvessel length. Bottom: Left: Representative confocal Z projection of microvessels at the beginning of culture. Right: Representative Z projection of initial microvessels in simulations. Depth of field for both images is 200 μm.
(TIF)

**S13 Fig. Schematic representation of simulated branching.** The direction of a new branch is determined by the direction of the parent microvessel, the zenith angle, and the azimuth angle. The zenith angle is the angle describing the elevation of the branch direction from the parent direction. The azimuth angle represents a rotation about the axis of the parent direction.
(TIF)

**S14 Fig. Emergent vessel tortuosity.** Z projections of microvascular networks grown in low (left) and high (right) anisotropy collagen using a continuous EFD method with the fibril weight $\alpha$ set to 0.9 (minimal persistence). Depth of field = 200 μm. High fibril weight leads to high curvature that resembles pathological angiogenesis.
(TIF)

**S15 Fig. Sensitivity study–Collagen fibril orientation weight ($\alpha$) and stress magnitude ($a_{amp}$).** Low-density results: The sensitivity of the model predictions to the neovessel sprout stress magnitude ($a_{amp}$) was studied across the full range of possible collagen orientation weights ($\alpha$). The ratio of the 2 greatest ODF semiprincipal axes was plotted for each condition on the y-axis. The results for 3 mg/mL simulations are displayed with results from discrete fibril vector simulations on the left and results from EFD simulations on the right. No differences are observable between the stress-free ($a_{amp}$ = 0.0 μPa) and baseline stress ($a_{amp}$ = 3.72 μPa). In contrast, the high-stress case led to increased polarization of the vascular network for a number of cases, as indicated by increased values on the y-axis. These effects are visibly more pronounced for the discrete simulations.
(TIF)

**S16 Fig. Sensitivity study–Collagen fibril orientation weight ($\alpha$) and stress magnitude ($a_{amp}$).** High-density results: The sensitivity of the model predictions to the neovessel sprout stress magnitude ($a_{amp}$) was studied across the full range of possible collagen orientation weights ($\alpha$). The ratio of the 2 greatest ODF semiprincipal axes was plotted for each condition on the y-axis. The results for 4 mg/mL simulations are displayed with results from discrete fibril vector simulations on the left and results from EFD simulations on the right. No differences are observable between the stress-free ($a_{amp}$ = 0.0 μPa) and baseline stress ($a_{amp}$ = 3.72 μPa). In contrast, the high-stress case led to increased polarization of the vascular network for a number of cases, as indicated by increased values on the y-axis. These effects are visibly more pronounced for the discrete simulations.
(TIF)

## Acknowledgments

We thank James B. Hoying and Hannah A. Strobel from Advanced Solutions Life Sciences LLC for their input and guidance for the predictive simulations.

## Author Contributions

**Conceptualization:** Steven A. LaBelle, Jeffrey A. Weiss.

**Data curation:** Steven A. LaBelle.

**Methodology:** Steven A. LaBelle, A. Marsh Poulson, IV, Adam Rauff, Gerard A. Ateshian, Jeffrey A. Weiss.

**Software:** Steven A. LaBelle, A. Marsh Poulson, IV, Steve A. Maas.

**Supervision:** Jeffrey A. Weiss.

**Validation:** Steven A. LaBelle, Gerard A. Ateshian.

**Visualization:** Steven A. LaBelle, Steve A. Maas, Adam Rauff.

**Writing – original draft:** Steven A. LaBelle.

**Writing – review & editing:** A. Marsh Poulson, IV, Adam Rauff, Gerard A. Ateshian, Jeffrey A. Weiss.

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
