## [Decision Letter · Decision Letter 0]

27 Apr 2023

Dear Professor Weiss,

Thank you very much for submitting your manuscript "Spatial Configurations of 3D Extracellular Matrix Collagen Density and Anisotropy Simultaneously Guide Angiogenesis" for consideration at PLOS Computational Biology.

As with all papers reviewed by the journal, your manuscript was reviewed by members of the editorial board and by several independent reviewers. In light of the reviews (below this email), we would like to invite the resubmission of a significantly-revised version that takes into account the reviewers' comments.

We cannot make any decision about publication until we have seen the revised manuscript and your response to the reviewers' comments. Your revised manuscript is also likely to be sent to reviewers for further evaluation.

Sincerely,

Philip K Maini

Academic Editor

PLOS Computational Biology

Jason Haugh

Section Editor

PLOS Computational Biology

Reviewer's Responses to Questions

**Comments to the Authors:**

Reviewer #1: This manuscript tackles a problem of current interest in mathematical modelling of angiogenesis, namely, how the growth of the vessel sprouts couples to the mechanics of the ECM. It is well known that such feedback (i.e. how the orientation of the fibers within the ECM guides the migration of ECs and the growth of the vessels, and how the vessels deform, realign and remodel the ECM) is crucial to understand the mechanisms of angiogenesis.

The authors present a methodology that allows them to quantify the anisotropy of the orientation distribution functions of the fibrils within the ECM. To do so they introduce the so-called ellipsoid fibril distribution (EDF) and an associated Monte-Carlo sampling method. The authors claim that this methodology allows them to improve the results obtained by means of previous approaches which introduced a vector field accounting for the average local orientation, since the latter does not contain enough information about the actual distribution of orientations.

While this is an interesting approach and the results look promising, the manuscript in its current form cannot be published and major revisions regarding the presentation of their model are required.

The major concern involves the description of the model for angiogenesis. While the description of how rate the growth is influenced by both concentration of collagen and fibril orientation seems adequate, information about how the fibril orientation feeds into the model of angiogenesis is lacking. In line 477, the authors mention that "The direction a tip grows depends on a combination of persistence (prior tip direction) and fibril orientation". This needs to be further explained and substantiated, specifically regarding the EFD description and the Monte-Carlo sampling by giving the corresponding formulae (as it has been done for the rate of growth in Eq. 11). I suggest that the authors write a section on model formulation which clearly states which compartments are considered in the model, how each of this compartments is actually modelled, and, more importantly, how the different compartments are coupled and integrated within the whole model. Without the information the accuracy/validity of claims such as "These differences arose due to the dependence of the growth rate on the local anisotropy; as neovessels grew up the positive gradient, the potency of matrix growth and guidance cues increased" (lines 263 to 265) cannot be assessed.

Another issue that should be taken into account is that the concept of the "orientation distribution function" is very informally defined. While the level of information provided is probably fine for a journal with a more specialised readership, a more detailed introduction to this concept should make the paper more accessible to a more general readership

Other issues that need clarification are:

1.- The acronym "SPD" does not seem to be defined

2.- Lines 214 to 216: "As a result, these simulations were

able to accurately predict anisotropic guidance in agreement with prior experimental measures. In

contrast, simulations using the vector field approach under-predicted anisotropic guidance." What is the significance of this "under-prediction"? For example, could the simpler model reproduce some of the features in Figs. 4b and 4c?

Reviewer #2: Uploaded as an attachment

Reviewer #3: I am sorry for this very short review due to time constraints. This an elegant approach to model the effect of ECM anisotropy on angiogenesis.

I have two questions/remarks:

- I am sorry if I missed it, but it seems that in this version there is only a one-way feedback of the matrix orientations on endothelial cell migration. Then external strains can lead to ECM remodeling. However, traction on the matrix by migrating endothelial cells would also remodel the matrix. Is this part of the model already, or could the authors comment on how this can be incorporated in the model in the future?

- At present it seems there is only qualitative comparison between models and experiment. Could a more quantitative comparison be provided? Also how does matrix orientation affect cell migration velocity in vitro and in silico?

**Have the authors made all data and (if applicable) computational code underlying the findings in their manuscript fully available?**

Reviewer #1: None

Reviewer #2: Yes

Reviewer #3: Yes

PLOS authors have the option to publish the peer review history of their article (what does this mean?). If published, this will include your full peer review and any attached files.

Reviewer #1: No

Reviewer #2: No

Reviewer #3: No
---

## [Decision Letter · Decision Letter 1]

4 Aug 2023

Dear Professor Weiss,

Thank you very much for submitting your manuscript "Spatial Configurations of 3D Extracellular Matrix Collagen Density and Anisotropy Simultaneously Guide Angiogenesis" for consideration at PLOS Computational Biology. As with all papers reviewed by the journal, your manuscript was reviewed by members of the editorial board and by several independent reviewers. The reviewers appreciated the attention to an important topic. Based on the reviews, we are likely to accept this manuscript for publication, providing that you modify the manuscript according to the review recommendations.

Sincerely,

Philip K Maini

Academic Editor

PLOS Computational Biology

Jason Haugh

Section Editor

PLOS Computational Biology

Reviewer's Responses to Questions

**Comments to the Authors:**

Reviewer #1: The authors have made a thorough revision of the paper in response to the reviewers comments

Reviewer #2: The review is uploaded as an attachment.

Reviewer #3: The authors have answered comment #2 on the quantitative comparison between model and experiment satisfactorily.

Regarding my comment #1 on the feedback between vessel tip traction and ECM reorientation, we get little insight as to what extent this aspect of the model affects the results. To what extent does it matter (in the model and also in the real system) that there is a cross-talk between vessel tip traction and ECM reorientation? In its present form the study seems to focus entirely on the effect of ECM structure on cell migration/angiogenesis. This is fine, but this limitation of the study should be clarified.

I would therefore recommend that the authors either:

(a) Include a new figure comparing the behavior of the model with and without vessel tip traction to give more insight into the extent that this important feature affects the model predictions;

or (b) Clarify the limitation of the study: i.e. discuss how it currently studies the effect of ECM structure on angiogenesis, and how it could be extended to get more insights in the two-way cross-talk between endothelial cells (ECs) and ECMs, and/or discuss if and why the effect of ECM structure on EC migration (one-way) is the most important.

**Have the authors made all data and (if applicable) computational code underlying the findings in their manuscript fully available?**

Reviewer #1: None

Reviewer #2: Yes

Reviewer #3: Yes

PLOS authors have the option to publish the peer review history of their article (what does this mean?). If published, this will include your full peer review and any attached files.

Reviewer #1: No

Reviewer #2: No

Reviewer #3: No

Figure Files:

Data Requirements:

Reproducibility:

References:

---

## [Decision Letter · Decision Letter 2]

29 Sep 2023

Dear Professor Weiss,

We are pleased to inform you that your manuscript 'Spatial Configurations of 3D Extracellular Matrix Collagen Density and Anisotropy Simultaneously Guide Angiogenesis' has been provisionally accepted for publication in PLOS Computational Biology.

Best regards,

Philip K Maini

Academic Editor

PLOS Computational Biology

Jason Haugh

Section Editor

PLOS Computational Biology

Reviewer's Responses to Questions

**Comments to the Authors:**

Reviewer #2: The authors have responded to my comments

Reviewer #3: The authors have satisfactorily addressed all my comments. I appreciate the insightful new analysis the feedback between vessel tip traction and ECM reorientation. Thank you. Great paper!

**Have the authors made all data and (if applicable) computational code underlying the findings in their manuscript fully available?**

Reviewer #2: Yes

Reviewer #3: None

PLOS authors have the option to publish the peer review history of their article (what does this mean?). If published, this will include your full peer review and any attached files.

Reviewer #2: No

Reviewer #3: **Yes: **Roeland Merks

---

## [Editor Report · Acceptance letter]

18 Oct 2023

PCOMPBIOL-D-23-00055R2 

Spatial Configurations of 3D Extracellular Matrix Collagen Density and Anisotropy Simultaneously Guide Angiogenesis

Dear Dr Weiss,

I am pleased to inform you that your manuscript has been formally accepted for publication in PLOS Computational Biology. Your manuscript is now with our production department and you will be notified of the publication date in due course.

With kind regards,

Anita Estes
